

# LSTM-based sentiment analysis for stock price forecast

Ching-Ru Ko[1] and Hsien-Tsung Chang[1,2,3,4]

[1] Department of Computer Science and Information Engineering, Chang Gung University, Taoyuan, Taiwan
[2] Bachelor Program in Artificial Intelligence, Chang Gung University, Taoyuan, Taiwan
[3] Department of Physical Medicine and Rehabilitation, Chang Gung Memorial Hospital, Taoyuan, Taiwan
[4] Artificial Intelligence Research Center, Chang Gung University, Taoyuan, Taiwan

## ABSTRACT

Investing in stocks is an important tool for modern people's financial management, and how to forecast stock prices has become an important issue. In recent years, deep learning methods have successfully solved many forecast problems. In this paper, we utilized multiple factors for the stock price forecast. The news articles and PTT forum discussions are taken as the fundamental analysis, and the stock historical transaction information is treated as technical analysis. The state-of-the-art natural language processing tool BERT are used to recognize the sentiments of text, and the long short term memory neural network (LSTM), which is good at analyzing time series data, is applied to forecast the stock price with stock historical transaction information and text sentiments. According to experimental results using our proposed models, the average root mean square error (RMSE ) has 12.05 accuracy improvement.

## INTRODUCTION

According to statistics from the Central Bank of Taiwan (*Central Bank of the Republic of China, 2021*), in the past two decades, the average annual fixed deposit interest rate has dropped from 5.02% to 0.77%. With the price index rising year by year, the wealth evaporates with inflation if people take conservative financial management. Therefore, more and more people will choose stocks with high returns and high liquidity as investment tools to make profits. It is not uncommon to see investment failures in the stock market. If one system can accurately forecast the stock trend, it will be a great help to investors.

Compared with other investment tools, the operating method for stocks bargaining is easy to understand. The amount of investment required is flexible, depending on the stock price of the investment target. According to the statistics (*TWSE, 2021*), we can notice that the number of stock investment account is approximately 10.64 million over 23 million population in Taiwan. It indicates that nearly half of Taiwanese have used or are using stocks as an investment tool. However, any investment is accompanied by considerable risks. In the past, in order to reduce risks and obtain higher benefits in the stock market, investors generally analyzed the fundamentals, technical aspects, and news of the target stocks. This paper focuses on analyzing relevant information on the news and forum articles. Because the price of a stock changes not only due the information from news, but

Corresponding author
Hsien-Tsung Chang,
smallpig@widelab.org

also the expected psychology and reaction of investors to this news from others. In view of the fact that news and forum articles are commonly used by the public investors to receive information of stock market. There are many recent studies showing that news (*Lee & Soo, 2017*) and forums (*Li, Bu & Wu, 2017*; *Liu et al., 2017*) will affect the price changing in the stock market.

For many years, whether in financial or academic aspects, stock price forecast has been a very important research topic. Many studies use statistical modeling or machine learning methods, such as Support Vector Machine (SVM), from historical data and then predict the future changes of the stock. In recent years, due to the advancement of GPU technologies, the computing power has been greatly improved. Related neural network models of deep learning have been accelerated and available for many successful applications in different fields along with a large amount of data training. This success helps us solve and deal with a lot of forecast applications. As we know, stock prices are time-series data, and it can be used for deriving patterns and identify trends using deep learning models. Perhaps these trends are too complicated to be understood by humans or other conventional computer processes. Recurrent Neural Network (RNN) is an artificial neural network suitable for solving time series problems. The connections between its neurons form a directed loop enabling it to act like dynamic time behaviors. RNN is now used in natural language processing, audio data processing, etc., and yields good results. The memory of original RNN will reduce its influence due to the number of complex layers after multiple recursions. Therefore, the long short term memory (LSTM) neural network concept concept is introduced. It is a special and important RNN model which can memorize long-term or short-term values and flexibly allow the neural network only to retain the necessary information. LSTM neural networks are suitable for the construction of stock price forecast models addressed by this paper.

With the latest advances in deep learning for natural language processing. More and more researchers use deep learning techniques to recognize the sentiments in the content of text or descriptive messages. One can instantly understand the stock investor's views on the stock market by recognizing the content of the forum posts and grasping the trend of stock investment. The performance of the stock market is also affected by news which is a public and global information source for everyone. By recognizing the sentiments in the news will become important vectors for predicting the stock prices. Therefore, this paper will try to combine stock price history and pre-training Bidirectional Encoder Representations from Transformers (BERT) model to recognize the sentiment as input vectors from news and forum posts for individual stock. Also, LSTM neural network models will be trained to forecast stock prices.

The main works of this study can be summarized as the following:

- We utilized multiple factors as fundamental analysis and technical analysis for the stock price forecast. This will be closer to the habits of ordinary stock investors for price forecasts.

- The news articles and forum posts from Taiwan famous PTT platform are both taken as the fundamental analysis. The state-of-the-art natural language processing tool BERT are used to recognize the sentiments of text.
- The long short term memory neural network (LSTM), which is good at analyzing time series data, is applied to forecast the stock price with stock historical transaction information and text sentiments.

## RELATED WORKS

Deep learning has become a very important research area for forecasting or recognition tasks in recent years. This section will describe researches that were related to LSTM neural networks, stock price forecast, text sentiment analysis, and the BERT model in the following.

### LSTM neural networks

The LSTM neural network is one of the derivatives of RNN. It not only improves the lack of long-term memory of RNN, but also prevents the problem of gradient disappearance. The LSTM neural network can dynamically learn and determine whether a certain output should be the next recursive input. Based on this mechanism that can retain important information, it provides a good reference and application when constructing a predictive model for this study.

The LSTM neural network has a new structure called memory cell (*Gao, Chai & Liu, 2017*). The memory cell contains four main components: input gate, Forget gate, Output gate and Neurons, through these three gates, decide what information to store and when to allow reading, writing and forgetting. Figure 1 illustrates how data flows through the storage unit and is controlled by each gate.

### Stock price forecast

In the past few decades, more and more people have conducted researches of stock prices forecast with the continuous expansion of the stock market. They try to analyze and predict fluctuations and price changes of the stock market. Stock prices are high dynamics, non-linearity, and high noise characteristics. The price of individual stock is affected by many factors, such as global economy, politics, government policies, natural or man-made disasters, investors' behavior, etc. It is one of the challenging tasks in time series forecasting problems.

*Li, Bu & Wu (2017)* and *Liu et al. (2017)* described that some stock market studies based on Random Walk Theory and Efficient Market Hypothesis. Those past studies believed that stock price fluctuations were random, so it cannot be predicted. According to the efficient market hypothesis, stock prices are driven by news, rather than current or past prices. Since news is unpredictable and stock prices will follow a random walk pattern. It is hard to forecast the stock price with the accuracy more than 50%. However, more and more researches in recent years have shown that the price of the stock market is not random and could be forecasted to some extent.

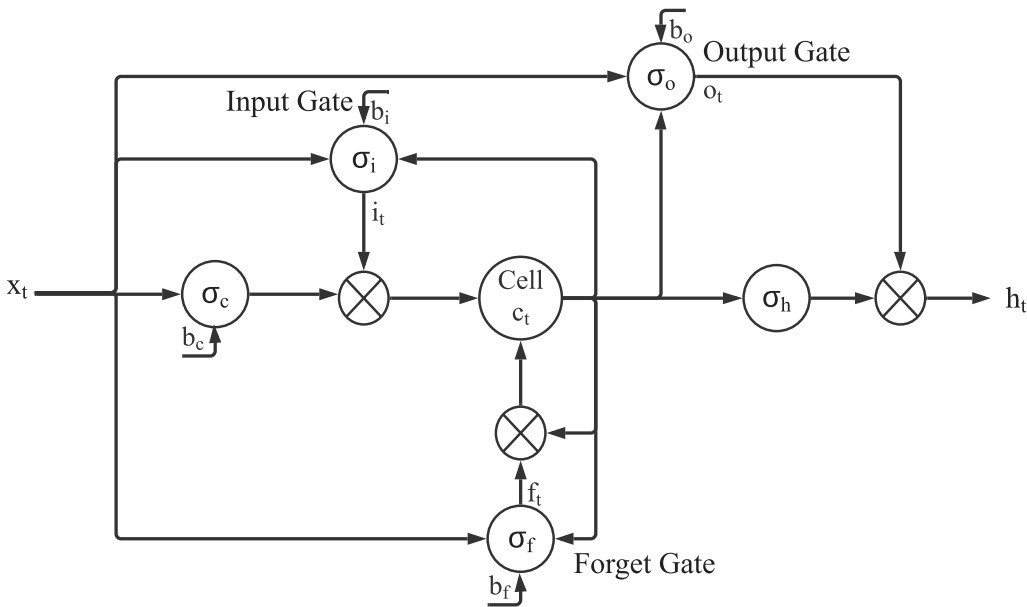

**Figure 1** Demonstration of the LSTM model *Gao, Chai & Liu (2017)*, where $x_t$ is the input vector in time $t$, $h_t$ is the output vector, $c_t$ stores the state of the union, $i_t$ is the vector of input gate, $f_t$ is the vector of forgotten gate, $o_t$ is the vector of output gate, and $\sigma_i$ $\sigma_f$ $\sigma_o$ $\sigma_c$ $\sigma_h$ are activation functions.

*Shah, Isah & Zulkernine (2019)* and *Selvin et al. (2017)* analyzed the stock market and predicted stock prices. They utilized two commonly used methods that were fundamental and technical analysis. Fundamental analysis is an investment analysis that estimates the value of stocks by analyzing company profiles, industry prospects, political factors, economic factors, news and social media. The information used in fundamental analysis is usually unstructured. This method is most suitable for long-term forecasting. The method of technical analysis tries to use the historical prices of stocks to predict the future development trend of individual stock. By record the daily rise and fall changes of the stock price in the form of charts. And then determine the best buying or selling points by observing its changes. K-line, Moving Average, and Relative Strength Index are commonly used algorithms in technical analysis and are suitable for short-term prediction.

The literature review of stock prediction *Shah, Isah & Zulkernine (2019)*; *Bustos & Pomares-Quimbaya (2020)* mentioned that technical analysis was one of the most commonly used methods to forecast the stock market and widely studied and used as a signal to indicate when to buy or sell stocks. However, some studies have found that returns obtained from the trading strategy are actually limited based on technical analysis. Fundamental analysis was rarely used in traditional researches because it is difficult to build models through relevant information. However, with the development of natural language processing and text analysis, some recent studies have analyzed non-structure stock related information to improve forecast accuracy. Those information could be official document, financial news, or posts on social networking sites. Due to the rapid progress of artificial intelligence, *Bustos & Pomares-Quimbaya (2020)* and *Nelson, Pereira & de Oliveira (2017)*

have achieved good results by using SVM and Artificial Neural Networks for stock market forecasts.

SVM is a tool of machine learning that can be used to deal with classification and regression problems. It is also called Support Vector Regression (SVR) for regression. It could identify the hyperplane in a high-dimensional feature space to accurately predict the distribution of data. *Xia, Liu & Chen (2013)* used SVR to create a stock forecast method based on regression model with historical time series data, today's opening price, highest price, lowest price, closing price, trading volume and adjusted closing price to predict opening price in the following day. The results have shown that the Mean Squared Error (MSE) is 0.0000253%.

RNN is often used among lots of ANN to deal with time series tasks, and it is also one of the most suitable techniques for dynamic time series forecasting. *Liu et al. (2017)* used RNN to predict stock volatility and found that the accuracy of the RNN model is better than MLP and SVM models. However, RNN will have the problem of gradient disappearance and explosion with multiple recursions. LSTM neural network has been developed to strengthen the operation of the RNN in the artificial intelligence fields. *Gao, Chai & Liu (2017)* collected the historical trading data of the Standard & Poor's 500 (S&P 500) from the stock market in the past 20 days as input variables, they were opening price, closing price, highest price, lowest price, adjusted price and transaction volume. They used LSTM neural network as the prediction model, and then evaluated the performance by Mean Absolute Error (MAE), Root Mean Square Error (RMSE), Mean Absolute Error Rate (MSE), and Mean Absolute Percentage Error (MAPE). Their method was better than other prediction ones. *Lee & Soo (2017)* combined LSTM neural network and CNN to predict the prices of Taiwanese stocks based on historical stock prices and financial news analysis, and found that it could reduce the error of prediction. *Li, Bu & Wu (2017)* used the LSTM neural network to predict the ups and downs of the China Securities Index 300 (CSI 300) by inputting the opening price, closing price and trading volume of the past ten days. The accuracy could reach 78.57%. *Khare et al. (2017)* also found that LSTM neural network can successfully predict the ups and downs of stock prices. *Lu (2018)* analyzed the positive and negative sentiment results only obtained from PTT posts and merged stock historical transaction information as the input vectors of LSTM neural network. Their proposed method can improve the accuracy of stock price prediction.

## Text sentiment analysis

Emotions or sentiments are biological states associated with the nervous system and reactions to external stimuli. They could be happiness, anger, sadness, fear, etc. We can divide them into positive or negative emotions by dichotomy. With the rapid development of the Internet and the popularization of mobile devices, people like to express opinions and exchange information through online platforms. It generates a large amount of text data containing emotional information, such as blog articles, social media, and online forum posts, replies, and product reviews, etc. If the emotions related to stocks expressed by users can be effectively analyzed from these text, it can help us understand the online public opinions in time.

According to researches *Li, Jin & Quan (2020)*, *Kumar & Garg (2019)* and *Ren, Wu & Liu (2018)* to text sentiment analysis (SA) are part of the field of natural language processing (NLP). A series of methods are proposed for online data like product review analysis, social network analysis, public opinion monitoring, movie box office forecasts, stock market fluctuation trend forecasts, etc (*Ali et al., 2019*; *Ali et al., 2020*; *Basiri et al., 2020*; *Li et al., 2020*).

*Mäntylä, Graziotin & Kuutila (2018)* expressed that the early Internet was not as developed and convenient as it is today. The amount of accumulated text data is not much, so there is less research on text sentiment analysis. Online comments, blogs, and social media (such as Facebook, Twitter) have emerged one after another after success of Web 2.0, and the number of text messages has increased rapidly. To process massive amounts of data, SA has become one of the most active research areas in NLP. *Turney (2002)* used an unsupervised algorithm to determine whether the review content of consumer review sites was recommended or not, including cars, banks. The average accuracy of the algorithm could reach 74%. *Pang, Lee & Vaithyanathan (2002)* adapted SVM and Na''ıve Bayes, and proposed Maximum Entropy Classification method to classify movie reviews into positive and negative emotional tendencies.

In recent years, many techniques have been proposed for sentiment analysis in different fields and tasks. *Li, Jin & Quan (2020)*, *Zainuddin, Selamat & Ibrahim (2018)*, and *Soong et al. (2019)* divide them into three categories. They are emotion calculation based on semantic dictionary, emotion classification method based on traditional machine learning and deep learning method.

The dictionary-based method does not require any training data. According to the open source emotional dictionary, count the emotional words that appear in the sentence. Each emotional word has an emotional score, and then calculate the emotional word score of the entire sentence. And it will output the emotional tendency of the sentence according to the score. Traditional machine learning methods use a large number of labelled corpora as training data to establish a classification model, and then apply this model to determine the emotional response of the target sentence, such as SVM, Naïve Bayes, and Decision Tree. In addition, since most emotional dictionaries have insufficient coverage of emotional words, lack of domain words, and ignoring context, and the performance of machine learning methods depends on the number of labeled samples, some studies have proposed a hybrid method combined with semantic dictionaries and machines. Learn to make up for each other's shortcomings to enhance the effectiveness of emotion analysis.

The study of *Bollen, Mao & Zeng (2011)* found that the text content of Twitter is related to the fluctuation of the Dow Jones Industrial Average Index (DJIA). *Liu et al. (2017)* analyzed the sentiments of the posts from the China Stock Forum and converted sentiments as the input of RNN to predict the volatility of the Chinese stock market. They found that sentiment indicators can effectively improve The accuracy of the forecast.

*Nofsinger (2001)* found that investor's sentiment is a key factor in the financial market. In some cases, investors tend to buy stocks after good news was released, and it will lead to rise stock prices. They sold stocks after negative news came out, so the price fell. Information

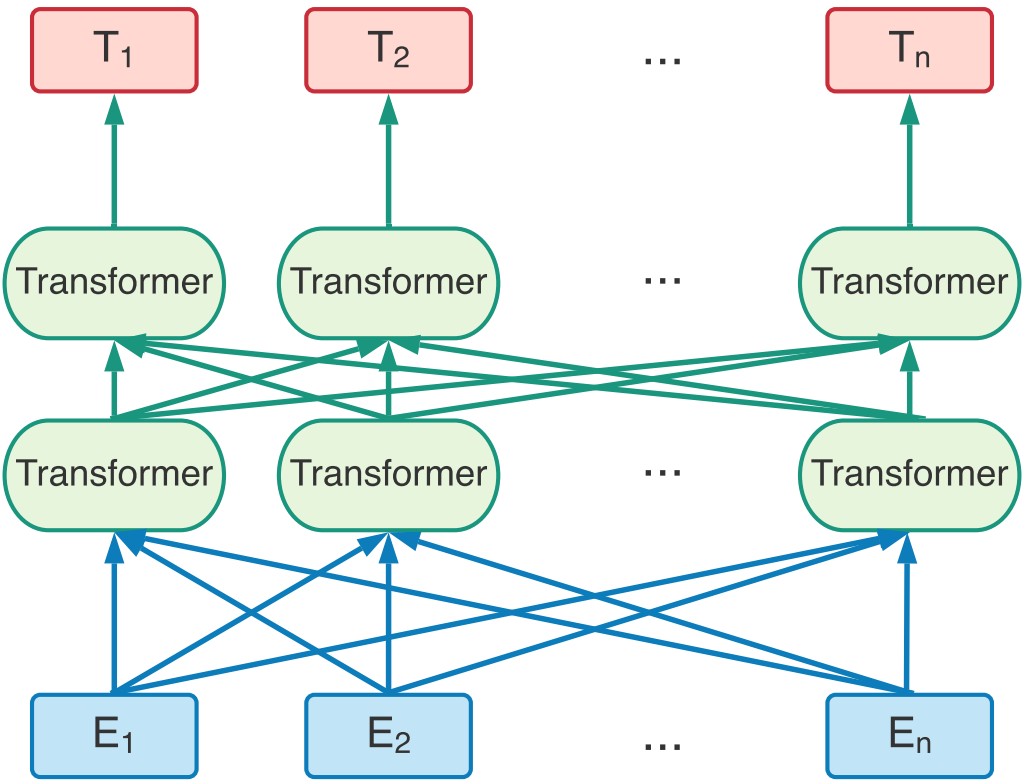

**Figure 2** The BERT pre-training model based on bi-direction transformer encoders. $E_1$ $E_2$ ..., $E_n$ are the input entities. $T_1$ $T_2$ ..., $T_n$ are the output from the BERT model (*Devlin et al., 2018*).

in the Internet provides valuable resources to reflect investor sentiments. Many researchers now use SA and news analysis to forecast stock prices.

## BERT

The structure of the BERT model is a multi-layer bidirectional transformer encoder. The transformer was a deep learning model using encoder and decoder for translation task. The BERT model took the advantages of the encoder part in transformer. Figure 2 is the BERT model diagram that takes $E_1$ $E_2$ ..., $E_n$ as inputs. They could be words or special symbols. After computed through multi-layer bidirectional transformer encoder, $E_1$ $E_2$ ..., $E_n$ are the output vectors. In the past researches, we used word2vec method try to map a word to a vector which was used the input of NLP. However, one word no matter in what context was always mapped to the same vector. The BERT model utilized multi-layer bidirectional transformer encoder, they will map a word into different vector according to different context around the input word. In other words, the BERT model is a model that will more precisely map a word to the word vector according the context. Unlike previous language representation models, BERT pre-training a deep bidirectional language representation model based on the upper and lower semantics of all layers.

After we received the word vector from the BERT model, we can combine other deep learning methods to solve NLP tasks utilized the benefits from word vectors that represents

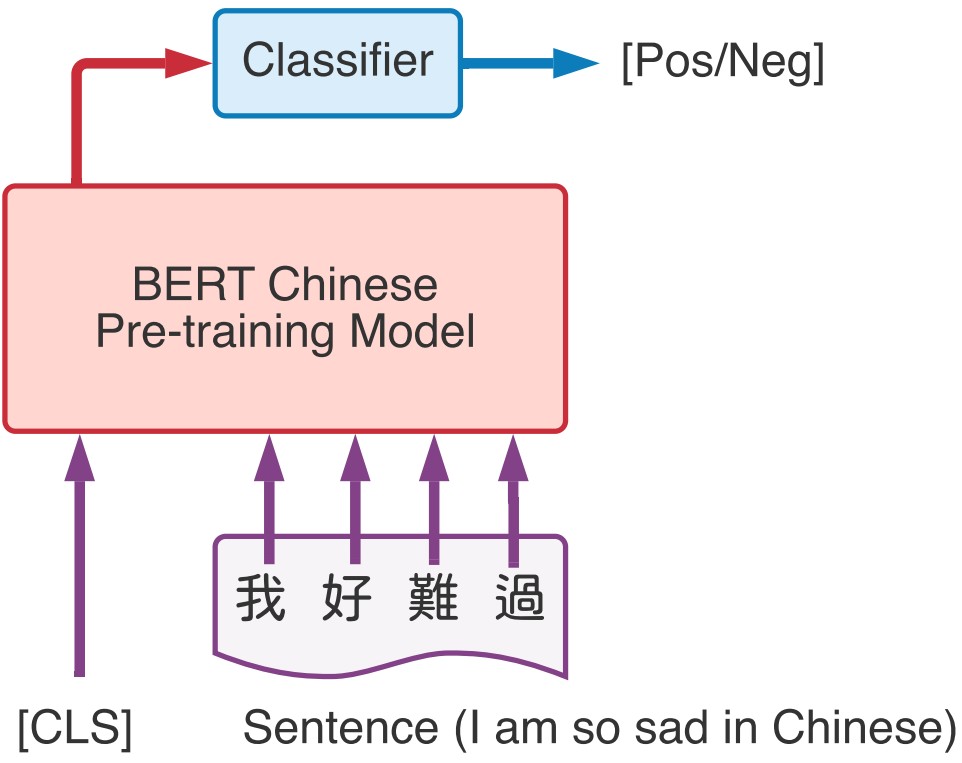

**Figure 3** Demonstration of BERT classification task.

the word according to different context. *Devlin et al. (2018)* implemented BERT through two main steps, namely pre-training and fine-tuning. Use a large amount of unlabeled data to pre-train the model for different tasks, and then use the labeled related data to fine-tune on specific downstream tasks (such as text classification, sentence-to-sentence classification, and question answering systems). Figure 3 is the demonstration of BERT fine-tuning process, taking the input of Chinese text and linear classifier to generate the output as an example. [CLS] is a special symbol added before each input. And the output word embedding vector from BERT pre-training model will be send to a linear classifier which can be used for subsequent classification tasks.

## METHODOLOGY

This section will explain the system architecture of our proposed method. It will describe the data collection and preprocessing, the BERT model, and the LSTM model for the stock price forecast in detail.

### System architecture

The stock market is affected by many factors. If you want to accurately predict the changes in the stock price of individual stocks, it is very important to effectively grasp the relevant information of the stock market. In this paper, we proposed a method that try to analyze

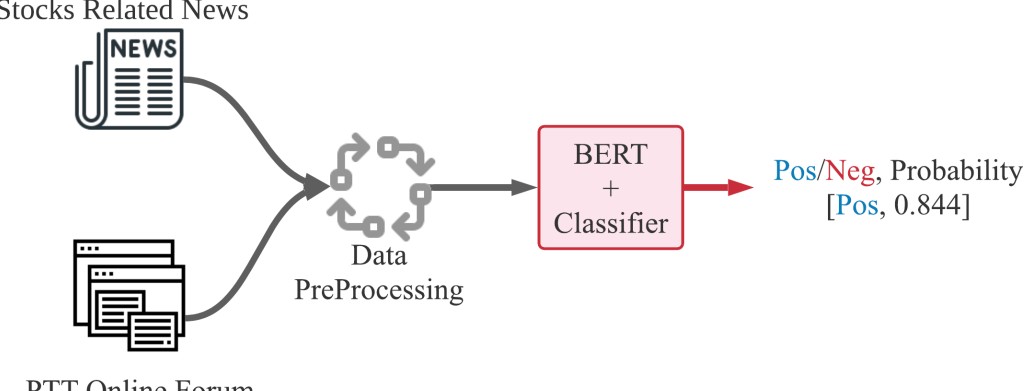

**Figure 4**  Text sentiment recognition process of sentences from news articles or forum posts.

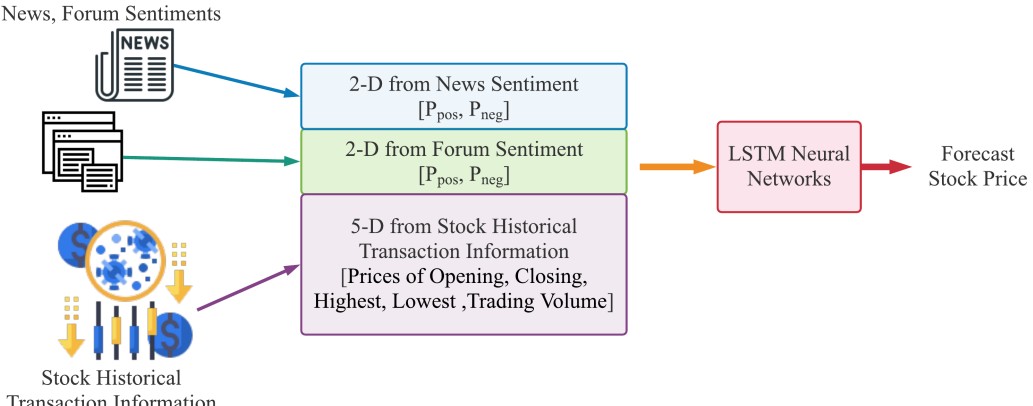

**Figure 5**  The data flow of our proposed method to forecast stock price using historical stock transaction information and both sentiments from news articles and forum posts, where $P_{pos}$ is the proportion of positive articles or posts and $P_{neg}$ is the proportion of negative ones.

sentiments in News and forum posts as the experience after fundamental analysis; and download the historical price of stocks as the technical analysis.

The data flow of our proposed method for stock price forecast is shown in Figs. 4 and 5. Figure 4 demonstrates the text sentiment analysis process for News and forum posts. The news was collected on the Internet and the forum posts were collected on the famous forum PTT stock board in Taiwan. Sentences extracted from articles or posts were sending into the BERT model to determine the positive or negative probabilities. Figure 5 shows that we combine four dimensions data of sentiments from news articles and forum posts along with five dimensions data of historical information, which are opening price, closing price, highest price, lowest price, and transaction volume in the past 20 days. The nine dimensions data are used as input features of the LSTM neural network prediction model to forecast the stock prices for individual stocks.

## Data collection

In this paper, we try to forecast stock prices. According to our proposed method, three categories of data are needed. They are historical stock transaction information, news, and forum posts from PTT. We mainly uses Python modules **Requests** and **Beautiful Soup** to perform web crawling and HTML parsing to obtain experimental data from January 1, 2015 to March 31, 2020.

First, we collect the daily transaction information of individual stocks from the open data from Taiwan Stock Exchange Corporation, including opening price, closing price, highest price, lowest price, and transaction volume. And then, we collect financial, political, and international news content related to individual stocks from *The Epoch Times (2021)*, *China-Times (2021)*, *Liberty Tines Net (2021)*, News in United Daily Network(UDN) (*United Daily Network, 2021*), Money Daily in UDN (*Money Daily, 2021*), *TVBS Media (2021)*, and *Yahoo (2021)*. Duplicated news is deleted.

PTT is one of the most famous and used online forums in Taiwan. A lot of users will posts or discuss stock market-related information on the stock board. We also collect posts related to individual stocks from PTT.

## Data preprocessing

The raw data of news articles and forum posts collected form Internet by crawler can not directly send to deep learning models. We will describe how we preprocessing the raw data in detail in the following subsections.

### *Preprocessing of news data*

After collected the news from Internet, we first keep all news related to individual stocks and separate the news into sentences. The accuracy of SA will be less according to the experiment if the the article is too long. In this paper, we will recognize the sentiments in an article according to sentences in it.

News is written in the form of formal articles, so there are certain rules for the use of punctuation in articles, usually a period, question mark, or exclamation mark. They are the end of a sentence, so the above three punctuation marks are used to segment the news content. After the sentences are segmented, we also filter out noises in sentences like special symbols, text in the note and braces. Sentences after preprocessed will be stored in the database for further sentiment analysis.

### *Preprocessing of forum posts from PTT*

The management of PTT stock board is rigorous, the board managers only allow posts within nine categories NEWS, EXPERIENCE, SUBJECT, QUESTION, INVESTMENT ADVICE, TALK, ANNOUNCE, QUESTIONNAIRES, and OTHERS. Figure 6 is the diagrammatic sketch of PTT forums' stock board. The categories of INVESTMENT ADVICE, TALK, ANNOUNCE, QUESTIONNAIRES, and OTHERS are not the related categories to this research, so they were excluded for further analysis. Each category of posts has detailed specifications, so the content of the article will be processed using different methods.

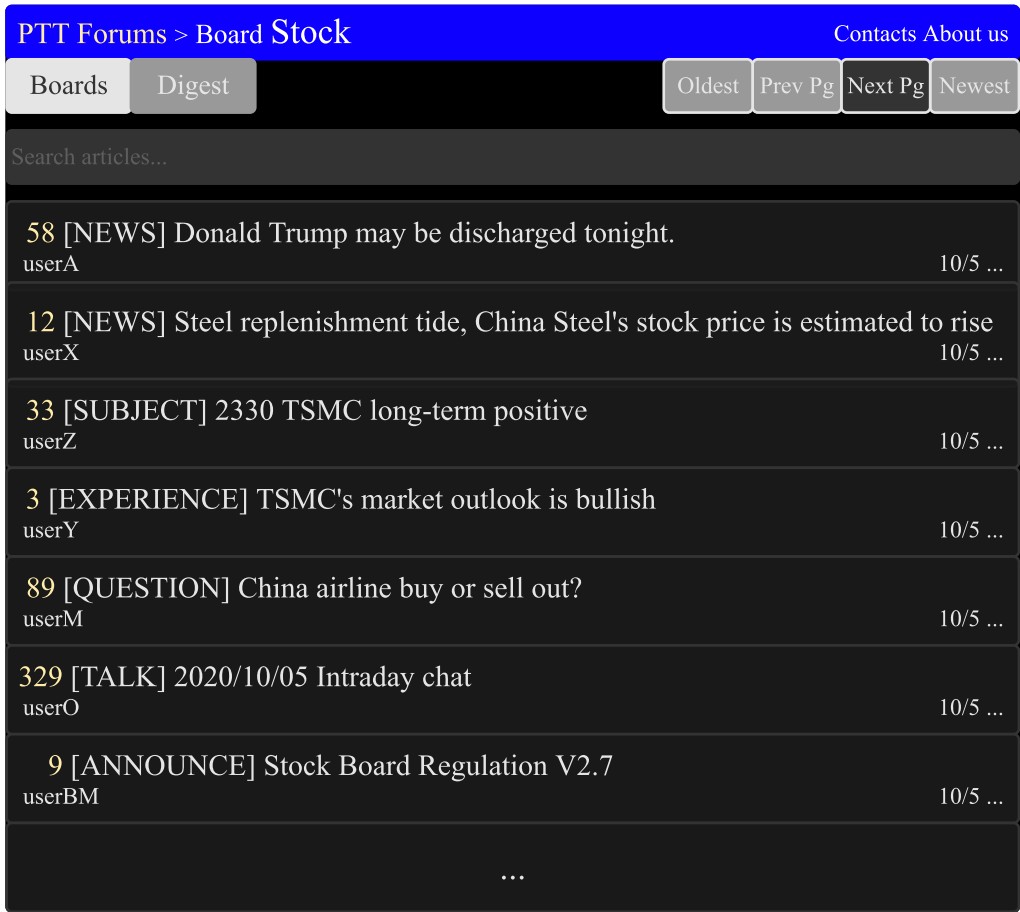

**Figure 6** **The diagrammatic sketch of the stock board in the PTT forum (*PTT-Forum, 2021*).** The title indicates the different categories of articles. [legend].

We will first search for posts that match the names of individual stocks from the post titles. And then remove special characters and URLs in the post or replies. Posts of the SUBJECT category can be divided into four sub-categories that is set by board managers. They are **Long** for suggestion to buy, **Short** for suggestion to sell, **Question** for asking a question, and **Experience** for expressing the experience to the subject stock. If it is **Long** or **Short**, it has been revealed that the author's sentiment towards this stock is positive or negative. The content of the article is only to explain his views and analysis, just label this post as positive or negative according to long or short. If the category is **Question**, it means that the author is only asking questions. The content of the post does not have emotions, you can omit it, just keep the replies. If it is **Experience**, the content of the article will be segmented by lines, because most authors end a sentence with a new line, and then sentences and replies are stored in the database.

The content of NEWS posts contains three items, which are the original link, the original content, and the replies of the posts. Because the original content is the same as the previously collected news, only the replies part is used for segmentation. Since the content

of the EXPERIENCE and QUESTION category posts is similar to the sub-category of the SUBJECT category, the same processing method is adopted. In addition, since the content of replies has a limit number of words, the content that exceeds the number of words will be automatically displayed in multiple replies. If the consecutive replies are written by the same user, they could be combined into one sentence. After the above processing, the sentence is stored in the database, and then sentiment analysis is performed.

## BERT model

The BERT code *Devlin (2019)* is open source provided by Google. We choose the traditional and simplified Chinese BERT pre-training model (Encoder with 12 layers of Transformer, 768 hidden units, 12 Self-attention heads, 110 million parameters). We apply the code on TensorFlow version 1.14.0 for text sentiment analysis.

9,600 sentences with manual labelled positive and negative sentiments are used as training data, 1,200 sentences are used as verification data, and 1,200 sentences are used as test data. A linear classification model after BERT is trained to perform classification. The fine-tuning parameters are set to the following values, the maximum length sequence is 300, the batch size is 16, the learning rate is 0.00002, and the epochs is 3.

Input the pre-processed sentences of news articles and PTT posts into our trained model, determine whether each sentence belongs to positive or negative sentiment. In order to avoid the result of sentiment classification affecting stock price prediction, only the sentence with probability equal or higher than 0.7 will be count as positive or negative. If the number of positive sentences in an article is more than negative, then this article will be regarded as a positive one, and vice versa. The calculation of Eqs. (1) and (2) will be used to obtain each positive and negative sentiment probabilities for news and PTT. Where $P_{pos}$ is the probability of positive articles or posts ($N_{pos}$) in $N_{total}$, and $P_{neg}$ is the probability of negative articles or posts ($N_{neg}$) in $N_{total}$. In this paper, we use dichotomy method for single article. There are a lot of articles related to one stock in a day, the values of $P_{pos}$ and $P_{neg}$ computed by Eqs. (1) and (2) can reduce the influence of error prediction in few articles.

$$P_{pos} = \frac{N_{pos}}{N_{total}} \tag{1}$$

$$P_{neg} = \frac{N_{neg}}{N_{total}} \tag{2}$$

## LSTM models for the stock price forecast

In order to recognize the impact of the sentimental information implicit in the news articles and PTT posts on the stock price changes, and to improve the accuracy of the forecast. We will select three stocks from top 50 volume-weighted stocks, and three stocks after top 50 from the open data of Taiwan Stock Exchange Corporation (*TWSE, 2021*). The selected stocks are based on the total number of news articles from online news sites (*The Epoch Times, 2021*; *China-Times, 2021*; *Liberty Tines Net, 2021*; *United Daily*

*Network, 2021*; *Money Daily, 2021*; *TVBS Media, 2021*; *Yahoo, 2021*) and PTT posts from online forum system (*PTT Forum, 2021*). They are Formosa Plastics Corporation (FPC 1301), Hon Hai Precision Industry Co., Ltd. (Hon Hai 2317), Taiwan Semiconductor Manufacturing Co., Ltd. (TSMC 2330), HTC Corporation (HTC 2498), China Airlines (CAL 2610), Genius Electronic Optical (GSEO 3406). The weight stocks are selected as the target of the forecast is mainly due to the amount of training data, because the weighted stocks are the stocks with larger equity and higher market value in Taiwan. Those companies usually have more news and discussion. In addition, we also want to know whether the experimental model can also improve the forecast results for individual stocks that account for a small proportion of the market. Therefore, from the stocks ranked below 50, three stocks are also selected for this research.

$$i_t = \sigma_i(W_i x_t + U_i c_{t-1} + b_i) \tag{3}$$

$$f_t = \sigma_f(W_f x_t + U_f c_{t-1} + b_f) \tag{4}$$

$$o_t = \sigma_o(W_o x_t + U_o c_{t-1} + b_o) \tag{5}$$

$$c_t = f_t \cdot c_{t-1} + i_t \cdot \sigma_c(W_c x_t + b_c) \tag{6}$$

$$h_t = o_t \cdot \sigma_h(c_t) \tag{7}$$

Use the historical transaction information of individual stocks from January 1, 2015 to March 31, 2020 (1279 trading days in total) which is downloaded from Taiwan Stock Exchange Corporation, news and PTT content with positive and negative sentiments as input vectors, and LSTM neural network are used for each input time series vectors. The model is used to forecast the price of multiple stocks. Figure 1 illustrates how data flows through the storage unit and is controlled by each gate in LSTM model. Eqs. (3), (4)– (7) express how to compute the values where $x_t$ is the input vector in time $t$, $h_t$ is the output vector, $c_t$ stores the state of the union, $i_t$ is the vector of input gate, $f_t$ is the vector of forgotten gate, $o_t$ is the vector of output gate, $W_i$, $W_f$, $W_o$, $W_c$, $U_i$, $U_f$, $U_o$ are weights, $b_i$, $b_f$, $b_o$, $b_c$ are the shift vectors, and $\sigma_i$ $\sigma_f$ $\sigma_o$ $\sigma_c$ $\sigma_h$ are activation functions.

Because the values range of historical trading data of the input data are not uniform, Min-Max Normalization will be performed before inputting into the LSTM neural network model. The value will be scaled to the interval from 0 to 1. As shown in Table 1, when forecasting is only through stock historical trading data, the RMSE value is mostly the smallest if the time steps is set to 20. This study makes forecasts based on stock prices in the past 20 days, as shown in Fig. 7. In the model, the neural network has three layers including an input layer, an LSTM layer and an output layer. Each nodes in the layer is connected to all nodes in the adjacent layer.

**Table 1  RMSE values of prediction stock price and true price from different stock in different time steps.**

| Stock ID | Time steps | | | | |
|---|---|---|---|---|---|
| | 5 | 10 | 15 | 20 | 25 |
| 1301 | 0.86 | 0.97 | 0.66 | 0.641 | 0.835 |
| 2317 | 1.846 | 1.976 | 1.438 | 1.222 | 0.831 |
| 2330 | 4.768 | 4.387 | 4.089 | 3.498 | 3.63 |
| 2610 | 0.189 | 0.207 | 0.098 | 0.073 | 0.074 |
| 2498 | 0.768 | 0.837 | 0.597 | 0.493 | 0.509 |
| 3406 | 11.329 | 10.946 | 10.246 | 6.97 | 6.013 |

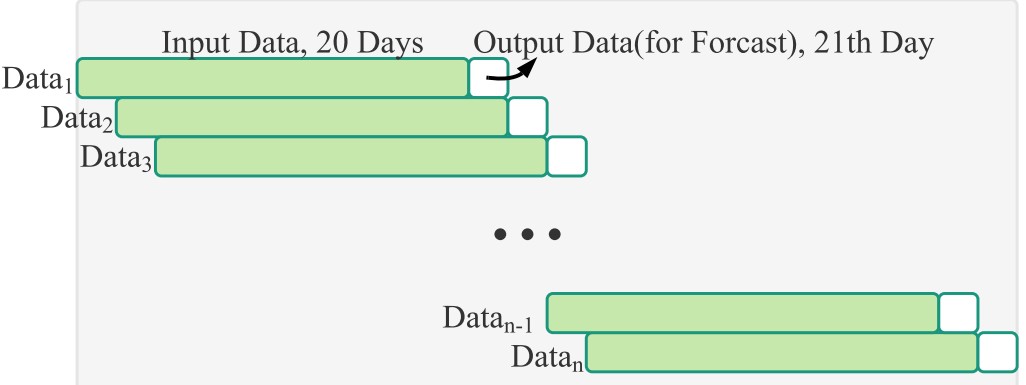

**Figure 7  Sliding window: input data of 20 days of stock transaction information and output data of the one day stock price.**

## EXPERIMENT AND RESULTS

In the past researches on stock price forecast based on machine learning only use stock historical trading data combined with the results of sentiment analysis of forum posts or news articles as input variables. However, we believe that news articles cover more long-term information and the content of the articles is more extensive. But the news updates are less immediate. Forum posts can reflect emergencies at any time, and may contain gossip. But they may contain incorrect information. In this study, we will simultaneously integrate news articles and forum posts as fundamental analysis vectors, and historical transaction data will become technical analysis vectors. The stocks price forecast will be conducted through the LSTM neural network.

The experiments were conducted as shown in Fig. 8. We first collect news and forum articles from web using crawlers, and download stock historical transaction information from Taiwan Stock Exchange Corporation. Those articles and information were stored in the database for further training and forecast. In the training stage, we first tag sentiment information for sentences and send to sentiment recognition classifier for training. The sentiment vectors from articles related to a specified stock will be send to LSTM neural

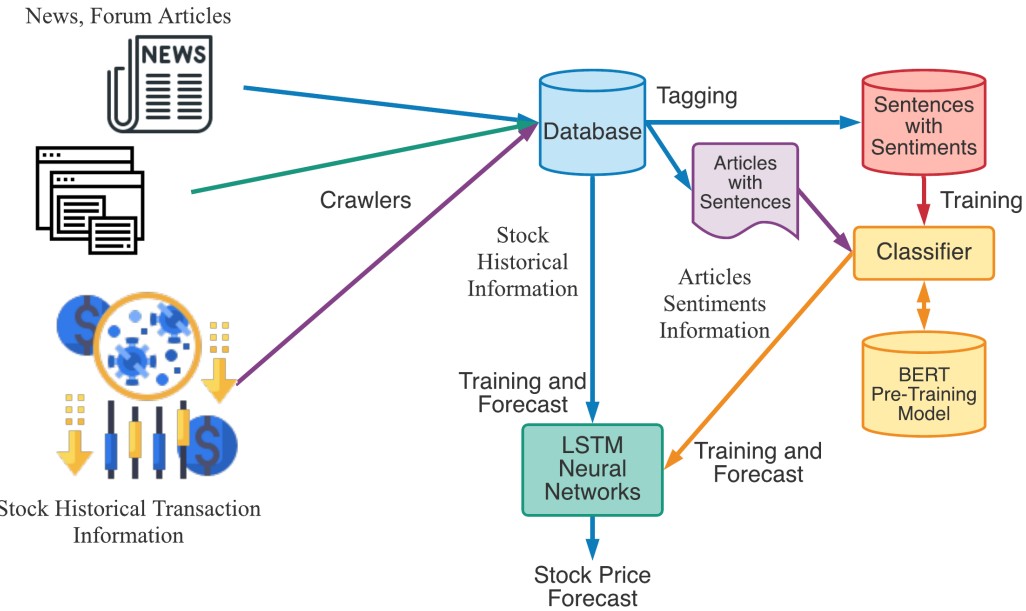

**Figure 8** **The experimental flow chart of our proposed stock price forecast model.**

**Table 2** **Method names and input vectors.**

| Names | Input Vectors |
|---|---|
| LSTMP | Opening price, Closing price, Highest price, Lowest price and Volume. |
| LSTMN | Opening price, Closing price, Highest price, Lowest price, Volume, $P_{pos}$ of news and $P_{neg}$ of news. |
| LSTMF | Opening price, Closing price, Highest price, Lowest price, Volume, $P_{pos}$ of PTT forum and $P_{neg}$ of PTT forum. |
| LSTMNF | Opening price, Closing price, Highest price, Lowest price, Volume, $P_{pos}$ of PTT forum, $P_{neg}$ of PTT forum, $P_{pos}$ of news and $P_{neg}$ of news. |

**Table 3** **RMSE values of prediction stock price and true price from different methods.**

| Method | Stock ID | | | | | |
|---|---|---|---|---|---|---|
| | 1301 | 2317 | 2330 | 2610 | 2498 | 3406 |
| LSTMP | 0.641 | 1.222 | 3.463 | 0.073 | 0.493 | 6.97 |
| LSTMN | 0.616 | 1.181 | 3.355 | 0.067 | 0.467 | 6.874 |
| LSTMF | 0.62 | 1.158 | 3.402 | 0.063 | 0.424 | 6.464 |
| LSTMNF | 0.602 | 1.108 | 3.307 | 0.058 | 0.411 | 5.874 |

networks along with stock historical transaction information for training. In the forecast

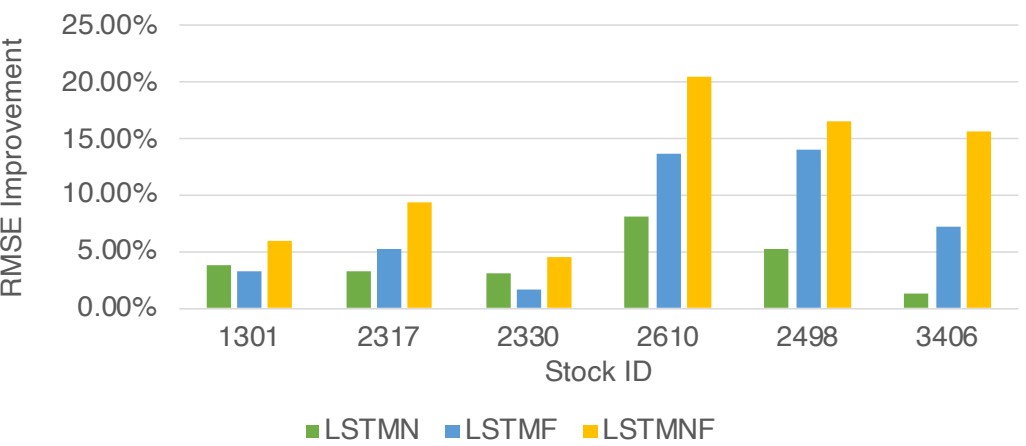

**Figure 9** $RMSE_{MethodImpr}$ **of different stocks and methods.**

stage, we will input the stock historical transaction information and articles sentiments information to trained LSTM neural networks for stock price forecast.

In this study, four different forecast models were established based on different input vectors. The details of input vectors and model names are shown in Table 2. The input vector will be send to LSTM neural networks for stock price forecast according to Fig. 8. The forecast results are evaluated with the RMSE which is calculated as shown in Eq. (8). The final experimental results are shown in Table 3. The improvement percentage of RMSE will be calculated as $RMSE_{MethodImpr.}$ in Eq. (9).

$$RMSE = \sqrt{\frac{\sum_{i=1}^{N}(Forecast_i - True_i)^2}{N}} \qquad (8)$$

$$RMSE_{MethodImpr.} = \frac{RMSE_{LSTMP} - RMSE_{Method}}{RMSE_{LSTMP}} \qquad (9)$$

In order to recognize the impact of the sentimental information implicit in the news articles and PTT posts on the stock price changes, and to improve the accuracy of the forecast. We select three stocks from top 50 volume-weighted stocks, they are also the constituent stocks in an important fund Taiwan 50, and three stocks after top 50 from the open data of Taiwan Stock Exchange Corporation (*TWSE, 2021*). We can notice that in Table 3 when the model only utilizes news articles (**LSTMN**) or PTT posts (**LSTMF**) sentiment analysis results, the RMSE value can be reduced. But the combination use of news and PTT sentiment analysis results makes the forecast results more accurate. As shown in Fig. 9, it shows the RMSE improvement of different stock for different methods. It can be seen that the experimental results have an average improvement of 12.05%. We have also drawn the actual forecast results into a graph, as shown in Fig. 10–15, and we can see that the forecast results are roughly in line with the trend of stock prices. In addition, Fig. 10–12 is one of the top 50 stocks in the weighted stocks, and Fig. 13–15 belongs to the

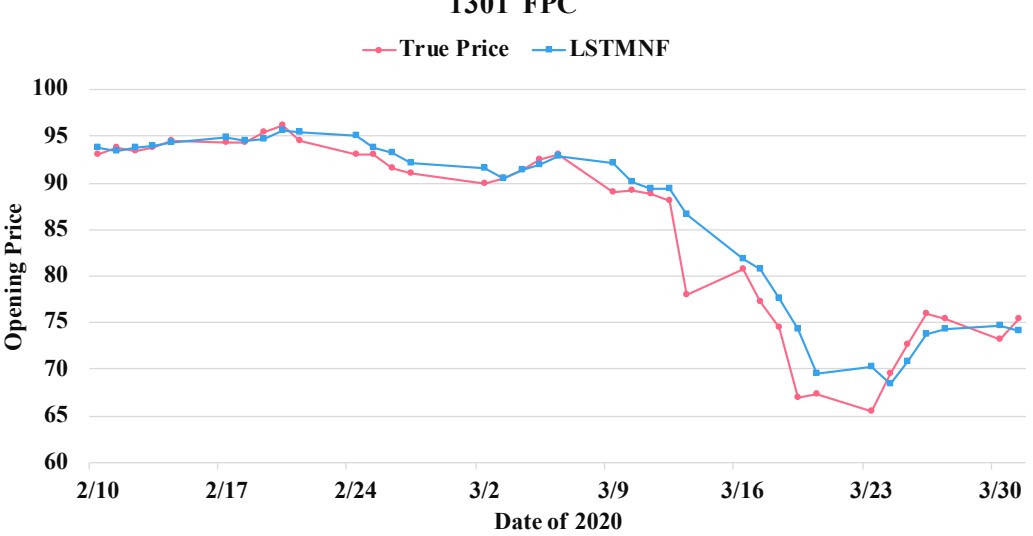

**Figure 10** Comparison of the forecast results of FPC (1301) in the true opening price and the LSTMNF model.

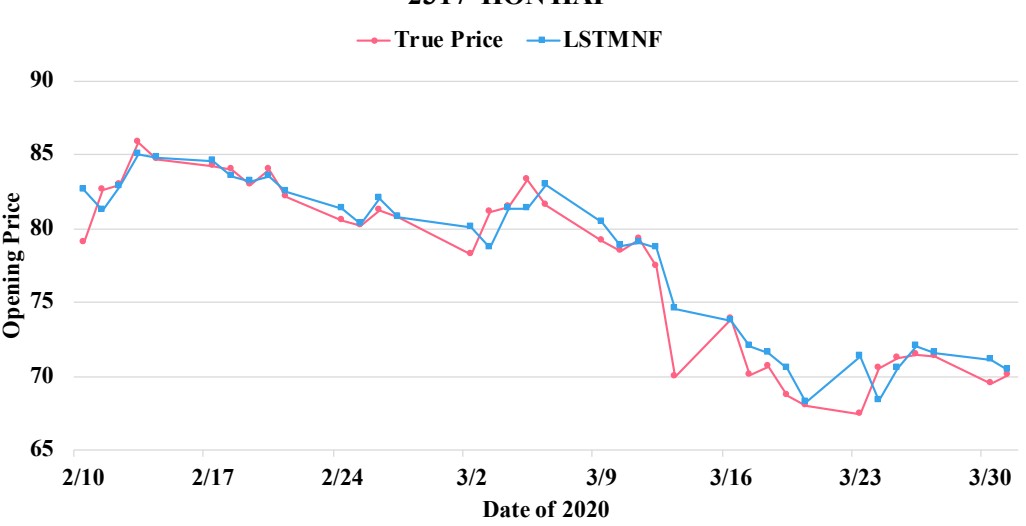

**Figure 11** Comparison of the forecast results of Hon Hai (2317) in the true opening price and the LSTMNF model.

stocks after the 50th in the weighted stocks. It shows that the model can also achieve good results for stocks that account for a small proportion of the market.

# CONCLUSION AND FUTURE WORK

This paper uses the pre-training language representation model BERT to perform sentiment analysis on the collected news articles and PTT forum posts related to individual stocks.

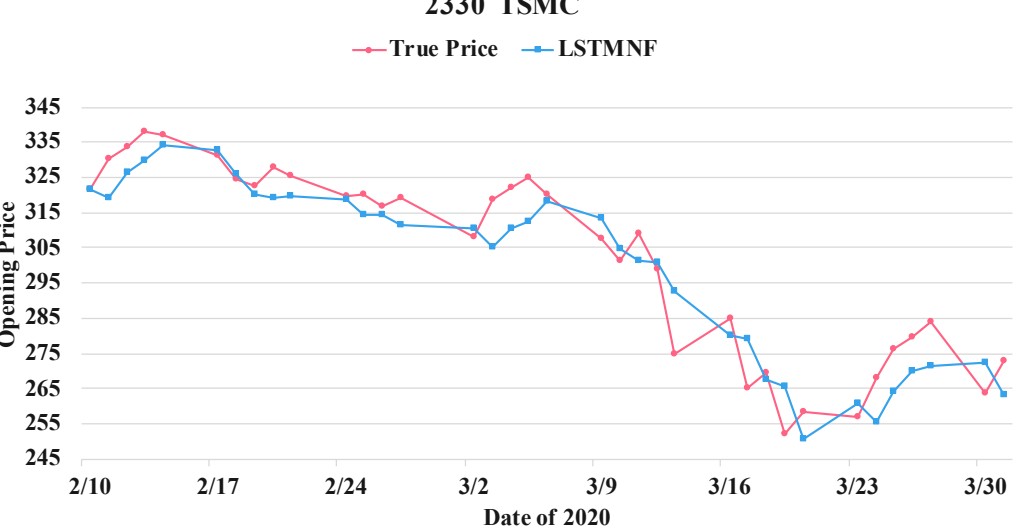

**Figure 12** Comparison of the forecast results of TSMC (2330) in the true opening price and the LSTMNF model.

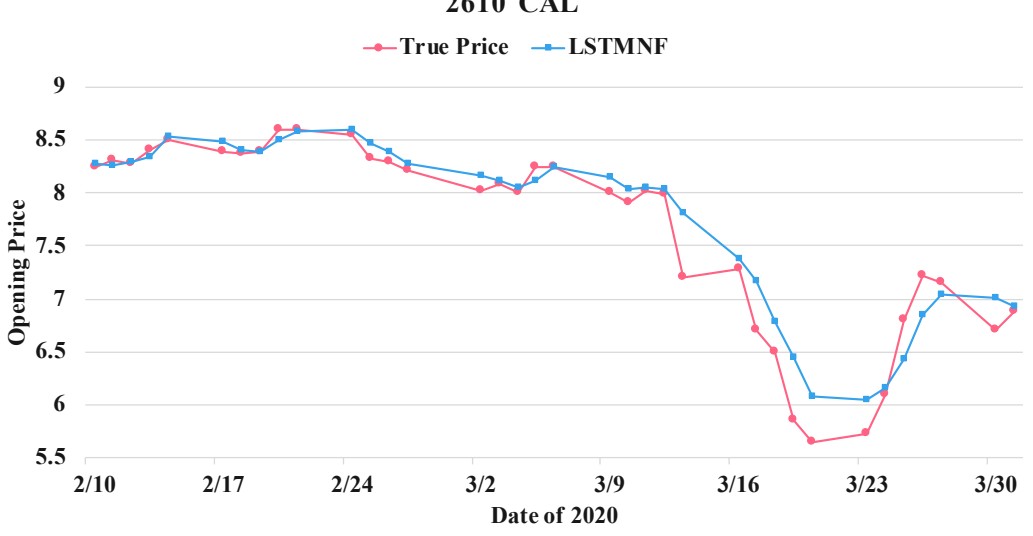

**Figure 13** Comparison of the forecast results of CAL (2610) in the true opening price and the LSTMNF model.

And we calculate the daily sentiment probabilities together with historical stock transaction data as input vector to LSTM neural network to forecast next day stocks opening price. The experimental results found that sentiment analysis features of news articles or PTT posts for the forecast model can reduce the RMSE. When we combine both sentimental information to the forecast model at the same time, the RMSE becomes smaller. Our proposed model LSTMNF in this study, an average improvement of 12.05% was obtained when compared

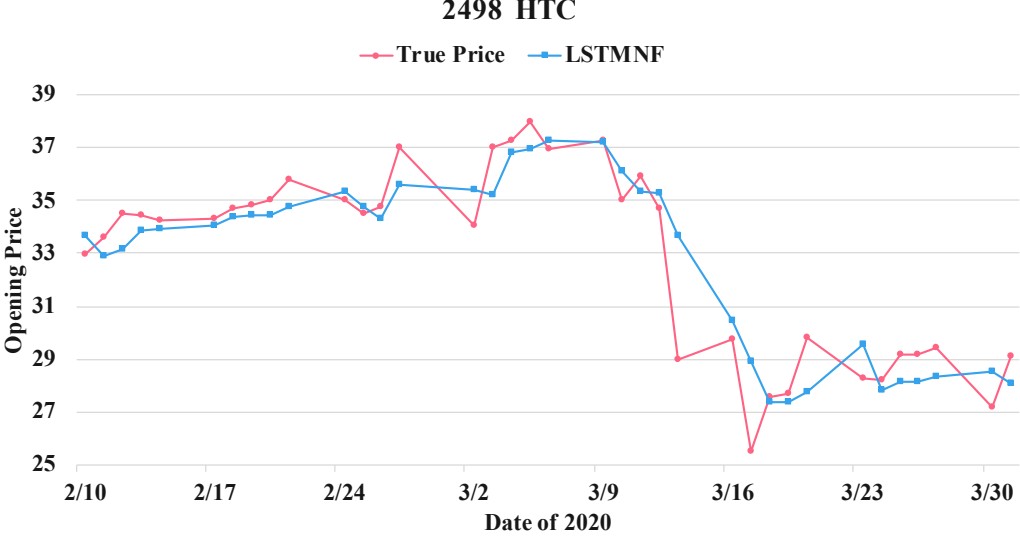

**Figure 14** Comparison of the forecast results of HTC (2498) in the true opening price and the LSTMNF model.

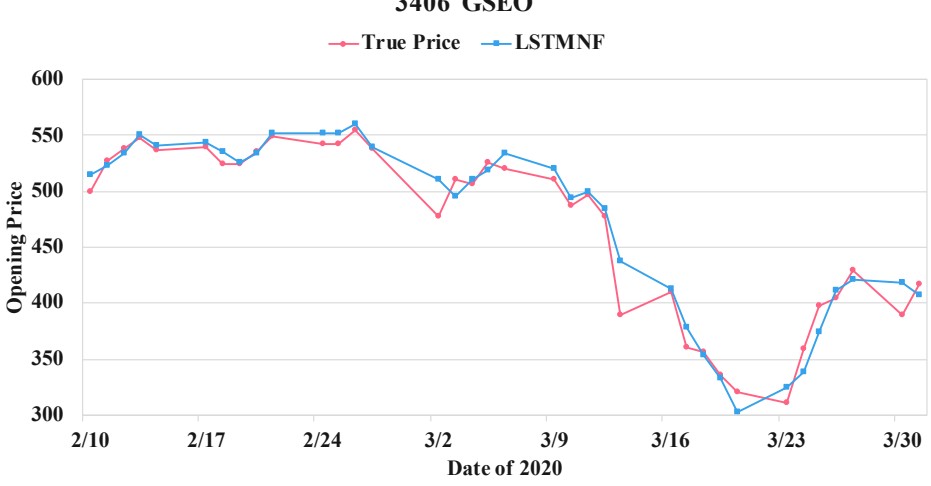

**Figure 15** Comparison of the forecast results of GSEO (3406) in the true opening price and the LSTMNF model.

to LSTMP. Therefore, it can be inferred that the sentiments implicit in news and forum play an important role in the stock market, which in turn affects the changes in stock prices.

Although the emotions in an article are simply separated into positive and negative in this paper, there are more emotion types could be recognized. Sad, fear, angry and disgust are all negative emotions, however the influence of stock price may be difference. If the emotion tag for an article could be labelled as an emotions vectors with the probability of

multiple types. The stock price could be more precisely predicted according to different types of emotions. This could be the clue for the future researches on this topic.

### Funding

This work was supported in part by the Ministry of Science and Technology, Taiwan, under Contract MOST 108-2221-E-182-049 (NERPD2J0281) and Chang Gung Memorial Hospital CMRPD2J0021. (Corresponding author: Hsien-Tsung Chang.) The funders had no role in study design, data collection and analysis, decision to publish, or preparation of the manuscript.

### Grant Disclosures

The following grant information was disclosed by the authors:
Ministry of Science and Technology, Taiwan: MOST 108-2221-E-182-049, NERPD2J0281.
Chang Gung Memorial Hospital: CMRPD2J0021.

### Competing Interests

The authors declare there are no competing interests.

### Author Contributions

- Ching-Ru Ko conceived and designed the experiments, performed the experiments, analyzed the data, performed the computation work, prepared figures and/or tables, authored or reviewed drafts of the paper, and approved the final draft.
- Hsien-Tsung Chang conceived and designed the experiments, analyzed the data, prepared figures and/or tables, authored or reviewed drafts of the paper, and approved the final draft.

### Data Availability

  The data and codes are available in Supplemental Files.

### Supplemental Information

Supplemental information for this article can be found online at http://dx.doi.org/10.7717/peerj-cs.408#supplemental-information.

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
