# Peer review of "LSTM-based sentiment analysis for stock price forecast"

_PeerJ Computer Science, doi:10.7717/peerj-cs.408_

## Round 0.1 · original submission · Major Revisions

I agree with the reviewers that major revisions are required. In particular, the presentation should be improved.

Reviewer 1 ·

Basic reporting

No comment.

Experimental design

No comment.

Validity of the findings

No comment.

Additional comments

The author uses machine learning and deep learning technology to predict the stock price, uses the Bert model to analyze the collected information's emotion, and inputs the analysis results and historical stock data into the LSTM neural network to predict the next day's stock price. However, many issues need to be addressed in this paper:
1. There are no keywords in the abstract. Please add keywords.
2. The background color of the abstract is not consistent with the whole article.
3. Variable B in Figure 1 is not represented in the figure; please, redraw.
4.The author ignores the influence of the error of emotion tags extracted by dichotomy on the whole prediction. Please explain why.
5. The description of the BERT method is too short, please introduce it in detail, and the structure diagram of the BERT model is not given, please add it.
6. The layout of all the pictures and tables is unreasonable, far away from the text description, and the impression is not good.
7. This paper lacks the figure of the model architecture and experimental flow-chart.
8. The sample stocks selected in this paper are less, and the experimental results are not universal.
9. The experimental results are not graphically represented, which is not intuitive enough.
10. If this is an innovative article, the model used is too old and lacks innovation.
11. The author designs four models with different inputs: LSTMP、LSTMN、LSTMF, and LSTMNF to compare the accuracy of forecasting stocks. Whether the four models have the same architecture, please give the architecture diagram. Only the experimental results of LSTMNF are given, and the other three models' experimental results need to be given.

·

Basic reporting

No comments

Experimental design

No comments

Validity of the findings

No Comments

Additional comments

In this paper, authors presented sentiment analysis approach based on LSTM Neural Networks for stock price forecast. However, I have some suggestion for paper improvement as follows.
1. The title of this manuscript should be changed. I suggest “LSTM-based sentiment analysis for stock price forecast”.
2. Authors should make the abstract more attractive.
3. The main contribution is not clear in introduction. Authors should discuss their main contribution in the form of bullets.
4. In related work Section, authors should include some sentences before starting subsection 2.1. In addition, authors should include existing research work about LSTM or Deep learning-based sentiment analysis (‘An intelligent healthcare monitoring framework using wearable sensors and social networking data’, Transportation sentiment analysis using word embedding and ontology-based topic modeling’, ‘ABCDM: An Attention-based Bidirectional CNN-RNN Deep Model for sentiment analysis’, ‘BiERU: Bidirectional Emotional Recurrent Unit for Conversational Sentiment Analysis’).
5. BERT should be discussed in detail.
6. Authors must include something before starting subsection (see 3.3 and 3.3.1).
7. I suggest to bring equations from section 2 (related work) to section 3.5 (LSTM models).
8. The future work should be included in conclusion.

---

## Round 0.2 · accepted · Accept

The issues raised by reviewers have been well addressed. I deeply appreciate the significant improvement that the authors have made.

·

Basic reporting

no comment

Experimental design

no comment

Validity of the findings

no comment

Additional comments

Thank you for addressing my comments.